# Annual Runoff Forecasting Based on Multi-Model Information Fusion and Residual Error Correction in the Ganjiang River Basin

**Peibing Song [1], Weifeng Liu [2], Jiahui Sun [3], Chao Wang [4],\*, Lingzhong Kong [5], Zhenxue Nong [6], Xiaohui Lei [4] and Hao Wang [1,4]**

1  College of Civil Engineering and Architecture, Zhejiang University, Hangzhou 310058, China; songpeibing@zju.edu.cn
2  College of Hydrology and Water Resources, Hohai University, Nanjing 210098, China
3  School of Civil Engineering, Shandong University, Jinan 250061, China
4  State Key Laboratory of Simulation and Regulation of Water Cycle in River Basin, China Institute of Water Resources and Hydropower Research, Beijing 100038, China
5  College of Hydraulic Science and Engineering, Yangzhou University, Yangzhou 225009, China
6  Guangxi Electric Power Design and Research Institute Co. Ltd., China Energy Construction Group, Nanning 530007, China
\*  Correspondence: wangchao@iwhr.com; Tel.: +86-10-6878-5503

**Abstract:** Accurate forecasting of annual runoff time series is of great significance for water resources planning and management. However, considering that the number of forecasting factors is numerous, a single forecasting model has certain limitations and a runoff time series consists of complex nonlinear and nonstationary characteristics, which make the runoff forecasting difficult. Aimed at improving the prediction accuracy of annual runoff time series, the principal components analysis (PCA) method is adopted to reduce the complexity of forecasting factors, and a modified coupling forecasting model based on multiple linear regression (MLR), back propagation neural network (BPNN), Elman neural network (ENN), and particle swarm optimization-support vector machine for regression (PSO-SVR) is proposed and applied in the Dongbei Hydrological Station in the Ganjiang River Basin. Firstly, from two conventional factors (i.e., rainfall, runoff) and 130 atmospheric circulation indexes (i.e., 88 atmospheric circulation indexes, 26 sea temperature indexes, 16 other indexes), principal components generated by linear mapping are screened as forecasting factors. Then, based on above forecasting factors, four forecasting models including MLR, BPNN, ENN, and PSO-SVR are developed to predict annual runoff time series. Subsequently, a coupling model composed of BPNN, ENN, and PSO-SVR is constructed by means of a multi-model information fusion taking three hydrological years (i.e., wet year, normal year, dry year) into consideration. Finally, according to residual error correction, a modified coupling forecasting model is introduced so as to further improve the accuracy of the predicted annual runoff time series in the verification period.

**Keywords:** annual runoff forecasting; factor selection; teleconnection factor; multi-model information fusion; residual error correction

## 1. Introduction

Hydrological forecasting, especially medium and long-term runoff forecasting, is an indispensable part of water resources management and water conservancy projects' operation [1–3]. Forecasting at different time scales can provide valuable information for flood control, power generation, water supply, and drought resistance [4–7]. Medium and long-term runoff forecasting, with a forecast period of

more than three days and less than one year, refers to scientific predictions of the future runoff before the occurrence of rainfall according to early hydrometeorological elements. In order to improve the accuracy and reliability of a runoff forecasting model, scholars at home and abroad have carried out massive application studies, in terms of selecting forecasting models and screening forecasting factors.

As far as forecasting models are concerned, cause analysis methods, mathematical statistics methods, and artificial intelligence methods proposed for improving the runoff prediction accuracy have received tremendous attention over the past decades [8,9]. Cause analysis methods pay attention to the physical formation process of hydrological phenomena, which comprehensively consider the influence of atmospheric circulation, meteorological factors, and the underlying surface physical environment on runoff variation. It is demonstrated that key hydrometeorological events, such as sunspot, EI Nino, ocean currents oscillation, and plateau snow, are closely related to runoff [10,11]. Nevertheless, cause analysis methods are mostly used for exploring the relationship between the atmospheric circulation and the hydrological elements, which are highly dependent on meteorological data and difficult to popularize. Time series methods and regression analysis methods are representative mathematical statistics methods that have been extensively adopted in runoff forecasting [12,13]. The former methods focus on the single-factor forecasting, while the latter methods place more emphasis on the multi-factor forecasting. Auto-regressive (AR), auto-regressive moving average (ARMA), auto regressive integrated moving average (ARIMA), and Markov chain methods have been general and popular time series models employed in hydrological forecasting [14,15]. Taking regression analysis methods as an example, key forecasting factors are screened from multiple forecasting factors that have a greater effect on the forecasting object on the basis of investigating the statistical rule between the forecasting factors and the forecasting object. As a whole, mathematical statistics methods avoid a mass of computation by taking some simple principles, but these methods have the disadvantages of low reliability and poor accuracy. It is also worth pointing out that the integrity and reliability of historical statistical data are equally important in mathematical statistics methods. Artificial intelligence methods, such as fuzzy mathematics, grey system, artificial neural network (ANN), and wavelet analysis, have the most applications in the current medium and long-term runoff forecasting. Mahabir et al. (2003) researched whether the fuzzy expert system was an alternative methodology for predicting the potential snowmelt runoff, and found that it was more reliable than the regression models in spring runoff forecasts, especially in terms of identifying low or average runoff years [16]. Trivedi et al. (2005) recommended that grey system theory may be a valuable tool for those watersheds possessing scanty hydrological data due to its uncertain mechanisms and insufficient information [17]. Compared to other intelligence methods, ANN has a wide application range in the hydrological fields because of its good robustness, strong nonlinear mapping and self-learning ability [18]. In spite of the good performance of these intelligence methods, there is still room to improve its prediction accuracy. With regard to ANN, there are certain differences in the results with each prediction for parameter uncertainty of neural network models. As a consequence, radial basis function (RBF), Elman neural network (ENN), adaptive neuro-fuzzy inference system (ANIFS), and long short-term memory (LSTM) are all alternative methods applied to predict runoff [19–22]. Moreover, in order to overcome the characteristics of complicated nonstationary runoff time series, empirical mode decomposition (EMD) and ensemble empirical mode decomposition (EEMD) proposed by Huang et al. (2003, 2008) have been new methods for nonstationary and nonlinear time series analysis [23,24]. In addition, hybrid models have been performed in many studies, because these models are capable of providing a high degree of accuracy and reliability compared to a single forecasting model [25]. Zhao et al. (2015) introduced a novel hybrid model made up of EEMD and AR for predicting nonstationary time series, and EEMD-AR was suitable for predicting the annual runoff of four hydrologic stations in the upper reaches of the Fenhe River basin [26]. A hybrid support vector machine–quantum behaved particle swarm optimization (SVM–QPSO) model was employed in predicting monthly streamflows, and it was able to deal with complex and highly nonlinear data

patterns. The prediction results indicated that the proposed hybrid model was a far better technique compared to the original support vector machine (SVM) model [27].

Apart from selecting appropriate forecasting models, identifying key predictors driving runoff variation is another step towards developing a reliable forecasting model [28]. Rough set (RS), global sensitivity analysis (GSA), factor analysis (FA), principal component analysis (PCA), Gamma test (GT), and forward selection (FS) techniques are used to reduce the number of input variables for recognizing forecasting factors [29]. With the development of information theory, mutual information (MI) as a measure representing information between two random variables provides optional means for screening forecasting factors [30]. To tackle key problems of generating minimal inference rule set and selecting complex factors, Zhu et al. (2009) proposed a forecasting model integrating the rough set theory with the fuzzy inference technique to improve the medium and long-term forecast precision [31]. Five principal impact factors were recognized by Li et al. (2012) by means of GSA and the back-propagation arithmetic, and these were pivotal factors that make a great difference to runoff during the flood season in the Nenjiang River Basin [29]. Some input selection techniques (e.g., GT, FS) designed to reduce the number of input variables, were fed to an SVM model to predict the monthly streamflow, and the developed GT-SVM model was superior to the original SVM model [32]. As a multivariate statistical technique used to identify important factors, PCA has been proposed to reduce the number of variables by providing a better interpretation of variables involving large volumes of information, as well as reducing the computational dimension [33,34]. Moreover, the information from independent and linear compound input variables is capable of presenting us with the minimum losses by employing this method [35]. Thus, PCA is acknowledged to be pivotal towards reducing the complexity of input variables and has been widely adopted into simplifying forecasting factors.

Nevertheless, it is simply not stable to rely on a single forecasting model, such as multiple linear regression (MLR), back propagation neural network (BPNN), Elman neural network (ENN), and particle swarm optimization-support vector machine for regression (PSO-SVR), to predict annual runoff, for runoff time series tending to be nonlinear, nonstationary and, even, chaotic. In view of this, multi-model information fusion technology and residual error correction methods are introduced to acquire more accurate annual runoff, taking the advantages of different forecasting methods into account [36,37]. The main objective of this paper is to develop a modified coupling forecasting model to predict annual runoff time series. Firstly, key forecasting factors are screened from two conventional factors (i.e., rainfall and runoff) and 130 atmospheric circulation indexes, with the help of PCA. Then, annual runoff time series are predicted by using the MLR model, the BPNN model, the ENN model, and the PSO-SVR model, respectively. Subsequently, a coupling model is constructed to predict annual runoff according to the multi-model information fusion technique. Finally, the residual error correction method is employed to further modify annual runoff time series in the validation period.

## 2. Study Area and Data Series

### 2.1. Study Area

As the seventh largest branch of the Yangtze River and the longest river of the Poyang Lake water system, the Ganjiang River Basin is located within 113°42′–116°38′ E and 24°30′–28°42′ N, Jiangxi province, China, controlling a drainage area of 83,500 km$^2$ and reaching a total length of 766 km [38]. With a subtropical humid monsoon climate, the Ganjiang River Basin has an annual average temperature ranging from 17 to 20 °C, an annual rainfall ranging from 800 to 1700 mm, and an annual mean inflow varying from 1300 to 3700 m$^3$/s. The Wan'an reservoir is located in the middle reaches of the Ganjiang River, 2 km upstream of the Wan'an County, with a drainage area of 36,900 km$^2$. As the largest water conservancy project in the Jiangxi Province, the Wan'an Reservoir plays a significant role in power generation, flood control, waterway transport, and irrigation.

However, since the water resources shortage, contradictions between water supply and demand have become frequent in the dry season in the middle and lower reaches of the Ganjiang River Basin

over the past years. Rational allocation of water resources is a key measure to solve above problems and minimize its adverse impact. Annual streamflow prediction is equally important for reservoir operation and water resource management in the Ganjiang River Basin, because it is the prerequisite and foundation for compiling a water resource allocation plan and carrying out reservoir operation. As an important control station in the upper reaches of the Ganjiang River Basin and a runoff monitoring station at the dam site of the Wan'an Reservoir, the Dongbei Hydrological Station is the basin outlet of the study area, controlling a drainage area of 40,231 km$^2$ [39]. Influenced by the plum rain, rainfall of the Dongbei Hydrological Station concentrates from March to June, accounting for about 54% of annual rainfall. Rainfall from July to September also occupies a large proportion of annual rainfall, due to the influence of the typhoon rain and the monsoon rain [40]. The location of the study area is shown in Figure 1.

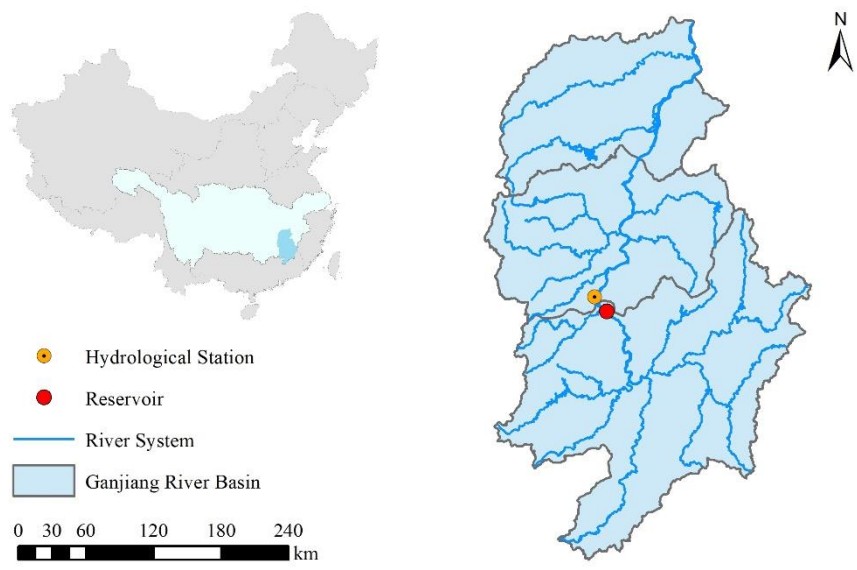

**Figure 1.** A brief description of the study area.

*2.2. Data Series*

Runoff and rainfall data of the Dongbei Hydrological Station covering 1964 to 2015 were obtained from the government hydrologic database. 130 atmospheric circulation indexes including 88 atmospheric circulation indexes, 26 sea temperature indexes, and 16 other indexes were provided by the National Climate Center of China Meteorological Administration (https://cmdp.ncc-cma.net/Monitoring/cn_index_130.php). Runoff data from 1964 to 2002 were used to confirm the forecasting model, while those from 2003 to 2015 were used to verify the forecasting model.

## 3. Research Methods

*3.1. Implementation of the Annual Runoff Forecasting*

The annual runoff forecasting model presented in this paper consists of four parts: extraction of key forecasting factors based on the principal component analysis (PCA) method; comparison of the predicted runoff time series of four forecasting models, including the multiple linear regression (MLR) model, the back propagation neural network (BPNN) model, the Elman neural network (ENN) model, and the particle swarm optimization-regression support vector machine (PSO-SVR) model; coupling three forecasting models (i.e., BPNN, ENN, PSO-SVR) by means of multi-model information fusion from the aspects of wet year, normal year, and dry year, respectively; modification of annual runoff time series predicted by the coupling model based on the residual error correction technique. In the

following sections, these parts will be elaborated upon in detail. The detailed technique flow chart is introduced in Figure 2.

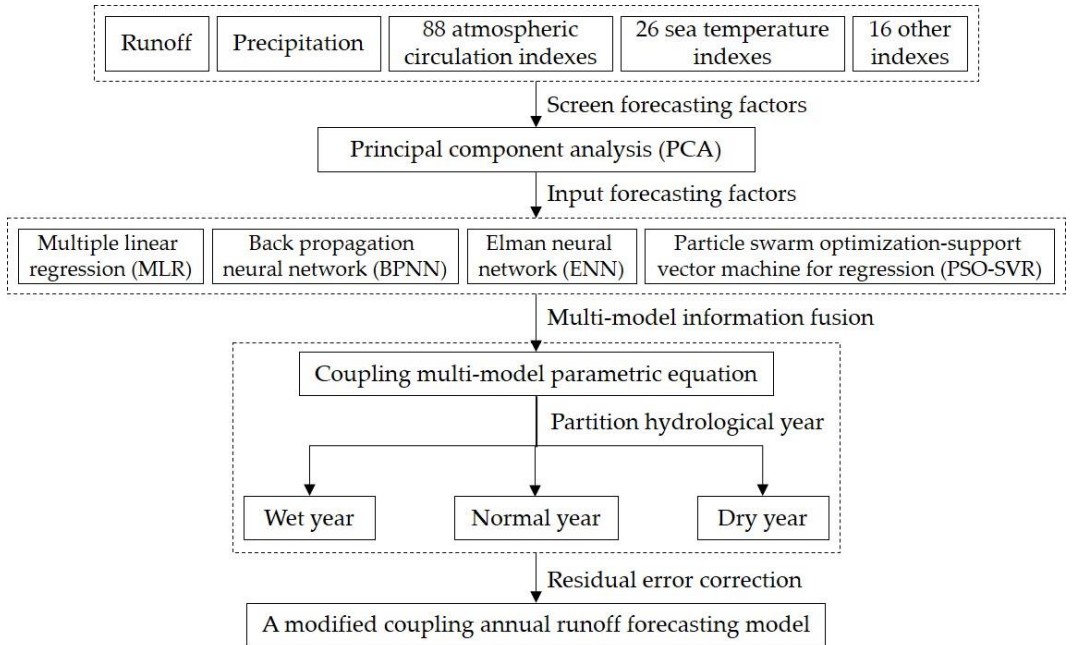

**Figure 2.** Technique flow chart of annual runoff forecasting.

## 3.2. Principal component analysis (PCA)

Principal component analysis (PCA) is a statistical technique aiming at reducing the dimensionality of a dataset with a large number of interrelated variables, while retaining most of the variability of the original datasets [41]. By calculating the cumulative contribution rate, PCA is applied to screen principal forecasting factors with an objective of losing as little information as possible. Given the initial forecasting factors $x_1, x_2, \cdots, x_n$ and the final forecasting factors $z_1, z_2, \cdots, z_m (m \leq n)$, the equation of extracting forecasting factors using PCA can be defined as follows:

$$
\begin{cases}
z_1 = l_{11}x_1 + l_{12}x_2 + \cdots + l_{1n}x_n \\
z_2 = l_{21}x_1 + l_{22}x_2 + \cdots + l_{2n}x_n \\
\cdots \\
z_m = l_{m1}x_1 + l_{m2}x_2 + \cdots + l_{mn}x_n
\end{cases} \tag{1}
$$

where $L$ is the load matrix composed of coefficient $l$. If $i \neq j$, $z_i$ has nothing to do with $z_j$. $z_1$ is the linear combination of $x_1, x_2, \cdots, x_n$, the variance of which is the largest among all linear combinations. In a similar way, $z_2$ is the linear combination of $x_1, x_2, \cdots, x_n$ that is not related to $z_1$, the variance of which is the second largest among all linear combinations. And so on.

## 3.3. Multi-Model Information Fusion

### 3.3.1. Multiple Linear Regression (MLR)

Multiple linear regression (MLR) is a classical statistical tool to describe the complex input-output relationship [42]. The key goal of MLR is to find out an approximation linear function between a set of independent variables and the dependent variable. Without a loss of generality, the regression line of MLR can be described as follows:

$$
y_i = \hat{y}_i + \varepsilon_i = \beta_0 + \beta_1 x_{1i} + \beta_2 x_{2i} + \cdots + \beta_k x_{ki} + \varepsilon_i (i = 1, 2, \cdots, n) \tag{2}
$$

where $k$ is the number of independent variables, $\beta_j (j = 1, 2, \cdots, k)$ is the partial regression coefficient, $x_i$ is the $i$ th independent variable, $y$ is the dependent variable, and $\varepsilon_i$ is the error term corresponding to $y_i$.

Then, the equation for a set of samples mentioned above can be rewritten in a compact matrix form, which can be expressed as follows:

$$Y = X\beta + \varepsilon \tag{3}$$

where $Y = \begin{bmatrix} y_1 \\ y_2 \\ y_3 \\ \cdots \\ y_n \end{bmatrix}_{n \times 1}$, $X = \begin{bmatrix} 1 & x_{11} & x_{21} & \cdots & x_{k1} \\ 1 & x_{12} & x_{22} & \cdots & x_{k2} \\ \vdots & \vdots & \vdots & & \vdots \\ 1 & x_{1n} & x_{2n} & \cdots & x_{kn} \end{bmatrix}_{n \times (k+1)}$, $\beta = \begin{bmatrix} \beta_0 \\ \beta_1 \\ \beta_2 \\ \vdots \\ \beta_k \end{bmatrix}_{(k+1) \times 1}$, $\varepsilon = \begin{bmatrix} \varepsilon_1 \\ \varepsilon_2 \\ \cdots \\ \varepsilon_n \end{bmatrix}_{n \times 1}$.

According to the classical matrix calculation theory, the least-square method can be adopted to calculate the coefficient vector $\beta$, and the coefficient vector can be described as follows:

$$\beta = (X^T X)^{-1} X^T Y. \tag{4}$$

In such a way, the coefficient vector $\beta$ is known, and the obtained MLR model can be used to predict the possible dependent variable related to the newly input vector.

### 3.3.2. Back Propagation Neural Network (BPNN)

The back propagation neural network (BPNN) is one of the most widely used neural network models, and it is a multi-layer feedforward network trained based on the error back propagation algorithm. BPNN can learn and store a large number of input–output mode mapping relations, and users can obtain relatively satisfactory prediction results without having to understand the mathematical equations of this mapping relation in advance [8]. BPNN continuously adjusts the weights and thresholds of the network through back propagation to achieve the least sum of square error. BPNN consists of three parts, namely an input layer, a hidden layer and an output layer, and its structure is drawn in Figure 3:

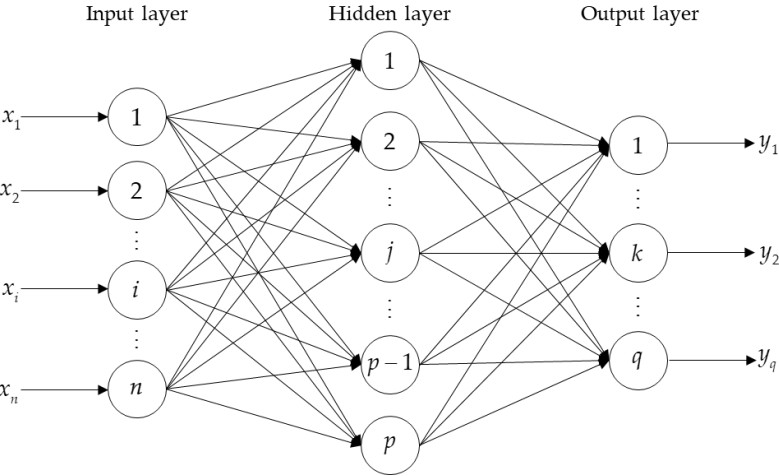

**Figure 3.** Structure of the back propagation neural network (BPNN).

where $X = (x_1, x_2, \cdots, x_i, \cdots, x_n)$, $HI = (hi_1, hi_2, \cdots, hi_j, \cdots, hi_p)$ and $YI = (yi_1, yi_2, \cdots, yi_k, \cdots, hi_q)$ are the input vector in the input layer, hidden layer, and output layer, respectively. $n$, $p$ and $q$ are the number of neurons in the input layer, hidden layer and output layer, respectively.

$HO = (ho_1, ho_2, \cdots, ho_j, \cdots, ho_p)$ and $YO = (yo_1, yo_2, \cdots, yo_k, \cdots, ho_q)$ are the output vector in the hidden layer and output layer, respectively. $DO = (do_1, do_2, \cdots, do_k, \cdots, do_q)$ is the expected output vector in the output layer. $w_{i,j}$ and $w_{j,k}$ are the input layer–hidden layer connection weight and the hidden layer–output layer connection weight, respectively. $BH = (bh_1, bh_2, \cdots, bh_j, \cdots, bh_p)$ and $BO = (bo_1, bo_2, \cdots, bo_k, \cdots, bo_q)$ are the threshold values corresponding to each neuron in the hidden layer and output layer, respectively. Given the number of samples and the activation function, the error function $e$ can be expressed as follows:

$$e = \frac{1}{2}\sum_{k=1}^{q}(do_k - yo_k)^2. \tag{5}$$

When BPNN is applied for predicting annual runoff time series, the corresponding output value is the prediction result on the premise of the specific input factors being transferred to the model. The learning procedures of the BPNN model are summarized as follows:

Step 1. Assign a random number to the connection weights between the layers, and determine the error function, as well as the given calculation error accuracy value and the maximum training time.

Step 2. Randomly select a sample as the input value and determine the expected output value.

Step 3. Calculate the input value and the output value of each neuron in the hidden layer.

Step 4. Calculate the partial derivative of the error function to each neuron in the output layer.

Step 5. Calculate the partial derivative of the error function to each neuron in the hidden layer.

Step 6. Correct the hidden layer–output layer connection weight.

Step 7. Correct the input layer–hidden layer connection weight.

Step 8. Calculate the global error and judge whether the model error meets the demand.

### 3.3.3. Elman Neural Network (ENN)

The Elman neural network (ENN) was firstly proposed by Elman (1990) to address the voice processing problem, and it is a typical dynamic recursive neural network. Based on the basic structure of BPNN, a context layer is added from the hidden layer to the input layer in the structure of ENN, and this context layer is taken as a one-step delay operator to record information from the last network iteration as input to the current iteration [43]. In addition, ENN enables the system to adapt to time-varying characteristics, as it enhances the global stability and has stronger computing power. A standard structure of ENN is drawn in Figure 4:

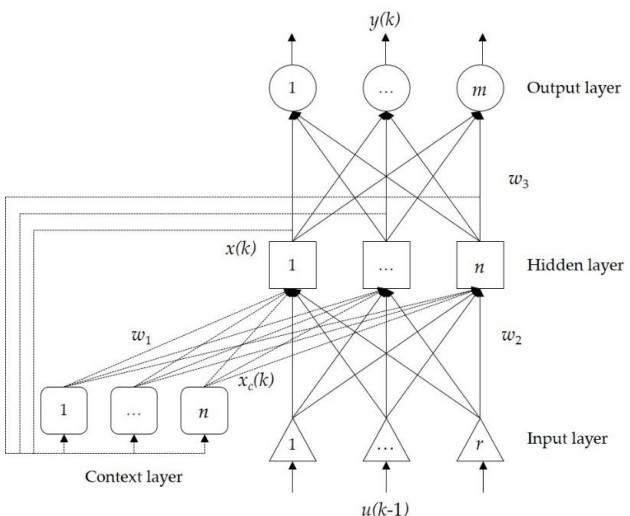

**Figure 4.** Structure of the Elman neural network (ENN).

where $u$ is the input vector, $y$ is the output vector, $x$ is the output vector in the hidden layer, $x_c$ is the output vector in the context layer. $w_1, w_2$ and $w_3$ denote the context layer–hidden layer connection weight, the input layer–hidden layer connection weight, and the hidden layer–output layer connection weight, respectively.

The state space of the ENN model can be expressed as follows:

$$\left\{ \begin{array}{l} y(k) = g(w_3 x(k)) \\ x(k) = f(w_1 x_c(k) + w_2 u(k-1)) \\ x_c(k) = x(k-1) \end{array} \right. \tag{6}$$

where $g(\cdot)$ is the transfer function of the neurons in the output layer, which is the linear combination of the hidden–layer output value. $f(\cdot)$ is the transfer function of the neurons in the hidden layer, which is commonly expressed by *S* function.

To achieve the minimum mean square deviation between the actual output value and the expected output value, the least-square algorithm and the gradient search technique are adopted in the ENN model. Then, the learning procedures of the ENN model are summarized as follows:

Step 1.　Normalize the original sample data. Set the maximum training time, the minimum expected error, the learning efficiency, and the sample number. The mean square error (MSE) function, is taken as the error function to describe the relation between the expected output value and the actual output value, and its equation can be expressed as follows:

$$MSE = \frac{1}{N} \sum_{t=1}^{k} \left( O_{ex}^t - O_{ac}^t \right)^2 \tag{7}$$

　　　　where $O_{ex}^t$ is the $t$ th expected output value. $O_{ac}^t$ is the $t$ th actual output value, and that is the observed value of the runoff.

Step 2.　Initialize the connection weights including $w_1, w_2$ and $w_3$. Train the ENN model for the first time when $t = 1$.

Step 3.　Calculate each input sample to find the output value in the hidden layer, the output value in the output layer, the sample error value, and the sample weight correction value. Calculate the global error value based on the sample error value.

Step 4.　Judge whether the global error is less than the specified accuracy. If it is, the ENN model ends its iteration and saves the current connection weight.

Step 5.　Judge whether the iteration time $t$ is less than the maximum iteration time. If not, the ENN model ends its iteration and saves the final connection weight.

Step 6.　Sum the sample weight correction value obtained in Step 3.

Step 7.　Correct the connection weight to obtain a new connection weight, taking the sample weight correction value obtained in Step 6 mentioned above into account. Turn to Step 3 mentioned above to continue to iterate.

### 3.3.4. Particle Swarm Optimization-Regression Support Vector Machine (PSO-SVR)

Support vector machine for regression (SVR) is a regression algorithm based on the support vector machine (SVM), and this technique has gradually become a new research hotspot in the fields of water resources engineering and hydrology [44,45]. The basic idea of the SVR model is to represent the entire sample set through a small number of support vectors and to map the training set to another high-dimensional feature space by means of nonlinear mapping. Then, the nonlinear function estimation problem in the input space can be transformed into a linear function estimation problem in the high-dimensional feature space. In spite of the efficiency of SVR for modeling nonlinear and complicated runoff time series, it still suffers from drawbacks, such as the selection of the parameters $C$, $\varepsilon$, and $\sigma$ making a great difference to the forecasting accuracy of an SVR

model [46]. Considering that the particle swarm optimization (PSO) algorithm has the advantages of easy implementation, fast convergence speed, and strong global search ability, the PSO algorithm is employed to optimize the free parameters of SVR. Moreover, applying the PSO algorithm into the parameter optimization of the SVR model has certain advantages compared to the traditional grid search methods [47]. The forecasting process of the PSO-SVR model is shown in Figure 5.

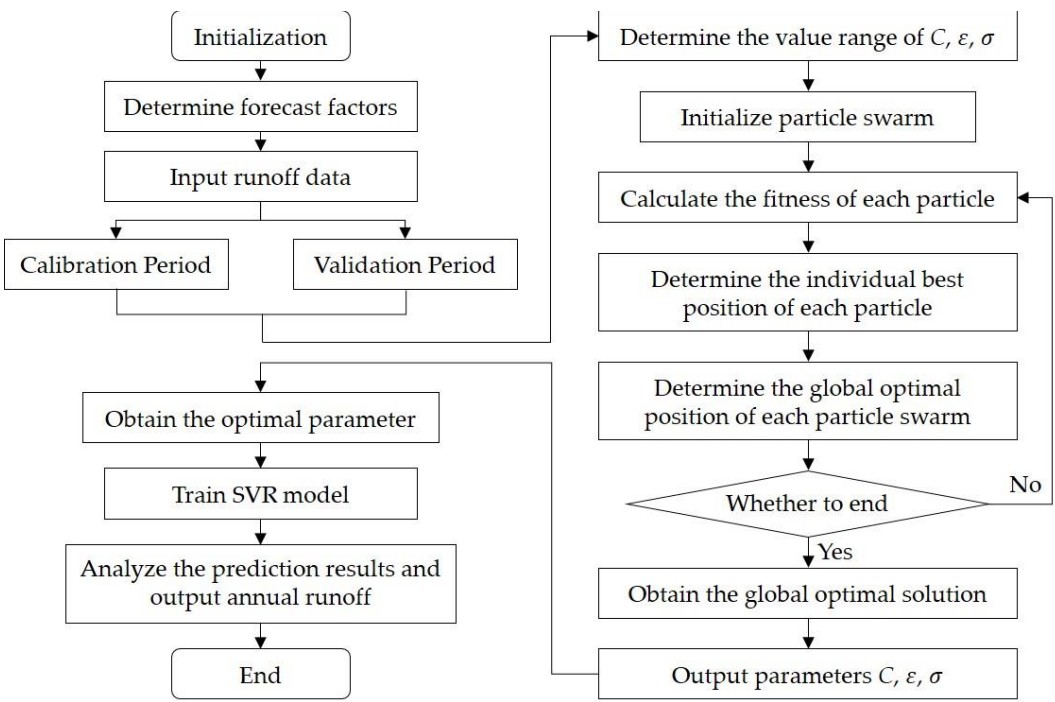

**Figure 5.** Forecasting process of the particle swarm optimization-regression support vector machine (PSO-SVR) model.

### 3.4. Residual Error Correction

Due to the high complexity and nonlinearity of runoff time series, as well as the instability of the forecasting model, it is not always ideal for the runoff forecasting results directly obtained by the forecasting model. Therefore, revising the forecasting results and improving the prediction accuracy is of necessity in the annual runoff forecasting. The simplest way to modify annual runoff sequences is to establish a regression error correction equation [36]. The specific steps of establishing a residual correction equation are as follows:

Step 1. This correction equation is established based on the predicted value and the corresponding residual value, and the residual sequences can be expressed as follows:

$$x_i = y_i - q_i \tag{8}$$

where $y_i, q_i$ and $x_i$ are the measured sequences, the predicted sequences, and the residual sequences, respectively.

Step 2. A residual equation between the predicted sequences and the residual sequences can be established as follows:

$$x_i = ay_i + b + \varepsilon_i (i = 1, 2, \cdots, n) \tag{9}$$

where $a, b$ are the regression coefficients, and $n$ is the number of sequences. $\varepsilon_i$ is the random error, and it obeys normal distribution $N \sim (0, \sigma^2)$.

Step 3. Regression coefficients $a, b$ are estimated by the least-square method, and the sum of the deviation square is calculated as follows:

$$S(a,b) = \sum_{i=1}^{n} (x_i - b - ay_i)^2. \tag{10}$$

In order to obtain the minimum $S(a,b)$, take the derivations of $S(a,b)$ to $a, b$, respectively, and define the derivative values as 0. The equations of the derivation process can be obtained as follows:

$$\begin{cases} nb + a\sum_{i=1}^{n} y_i = \sum_{i=1}^{n} x_i \\ b\sum_{i=1}^{n} y_i + a\sum_{i=1}^{n} y_i^2 = \sum_{i=1}^{n} x_i y_i \end{cases}. \tag{11}$$

The equations mentioned above are normal equations, and the solution is not the true values of the regression coefficients $a, b$ but the estimated values. Then, replace the true values $a, b$ with the estimated values $\hat{a}, \hat{b}$, and above equations can be rewritten as follows:

$$\begin{cases} n\hat{b} + \hat{a}\sum_{i=1}^{n} y_i = \sum_{i=1}^{n} x_i \\ \hat{b}\sum_{i=1}^{n} y_i + \hat{a}\sum_{i=1}^{n} y_i^2 = \sum_{i=1}^{n} x_i y_i \end{cases}. \tag{12}$$

In this case, the regression coefficients have a unique solution:

$$\begin{cases} \hat{a} = \dfrac{\sum_{i=1}^{n}(x_i - \bar{x})(y_i - \bar{y})}{\sum_{i=1}^{n}(y_i - \bar{y})^2} \\ \hat{b} = \bar{x} - a\bar{y} \end{cases}. \tag{13}$$

Calculate the regression equation between $y$ and $x$:

$$x_i = \hat{a}y_i + \hat{b}(i = 1, 2, \cdots, n). \tag{14}$$

Step 4. Calculate the residual values corresponding to the predicted sequences.
Step 5. Calculate the modified prediction values by the predicted residual values:

$$\hat{y}_i = y_i - x_i. \tag{15}$$

### 3.5. Evaluation Index

In order to evaluate the performance of the annual runoff forecasting model, three main criteria including mean absolute relative error (MARE), absolute relative error (ARE), root mean squared error (RMSE), qualification rate (QR) and Nash–Sutcliffe efficiency coefficient (NS) are taken as evaluation indexes, and these criteria are suitable to describe the prediction accuracy [19]. MARE and ARE are employed for examining the error between the predicted data and the observed data [48]. RMSE is chosen as an evaluation index to test the differences between the predicted data and the observed data [49]. In the standard for hydrological information and hydrological forecasting, QR is an important index to distinguish the accuracy grade of runoff projects [50]. NS is a popular evaluation index for evaluating the performance of forecasting models [51].

(1) The equation of mean absolute relative error (MARE) and absolute relative error (ARE) can be expressed as follows:

$$MAPE = \frac{1}{k}\sum_{i=1}^{k}\left|\frac{Q_{sim,i} - Q_{obs,i}}{Q_{obs,i}}\right|, \ ARE = \left|\frac{Q_{sim,i} - Q_{obs,i}}{Q_{obs,i}}\right| \tag{16}$$

where $Q_{sim,i}$ is the predicted value of the $i$ th sample, $Q_{obs,i}$ is the observed value of the $i$ th sample, and $k$ is the number of samples.

(2) The equation of root mean squared error (RMSE) can be expressed as follows:

$$RMSE = \sqrt{\frac{1}{k}\sum_{i=1}^{k}(Q_{sim,i} - Q_{obs,i})^2}. \tag{17}$$

(3) The equation of qualified rate (QR) can be expressed as follows:

$$QR = \frac{m}{n} \times 100\% \tag{18}$$

where $m$ is the time of the qualified forecasts, and $n$ is the time of the total forecasts. If the absolute value of the relative error between the predicted value and the measured value is within 20% the forecast is qualified.

(4) The equation of Nash–Sutcliffe efficiency coefficient (NS) can be expressed as follows:

$$NS = 1 - \frac{\sum_{i=1}^{k}(Q_{obs,i} - Q_{sim,i})^2}{\sum_{i=1}^{k}(Q_{obs,i} - \overline{Q}_{obs})^2} \tag{19}$$

where $\overline{Q}_{obs}$ denotes the average value of the observed annual runoff time series.

## 4. Results

### 4.1. Determining Forecasting Factors

Based on annual runoff time series from 1964 to 2015 of the Dongbei Hydrological Station, annual rainfall time series from 1963 to 2015, and 130 monitoring indexes (i.e., 88 atmospheric circulation indexes, 26 sea temperature indexes, 16 other indexes) from 1963 to 2014, the forecasting factors are screened according to the principle of selecting principal components whose cumulative contribution rate is greater than 85%. When analyzing the correlation between the forecasting factors of the year before the forecast year and the runoff sequence of the forecast year, the cumulative rainfall of the year before the forecast year is also added as the forecasting factor by the PCA method for dimensionality reduction, considering rainfall sequence is a key factor affecting annual runoff variation. The variance contribution-rate ranking of components is shown in Table 1.

**Table 1.** Variance contribution-rate ranking of components (note: units of the variance contribution rate and the cumulative variance contribution rate are %).

| Component | Initial Eigenvalue | | | Extraction Sum of Squared Loading | | |
|---|---|---|---|---|---|---|
| | Total | Variance | Cumulative Variance | Total | Variance | Cumulative Variance |
| 1 | 4.02 | 30.93 | 30.93 | 4.02 | 30.93 | 30.93 |
| 2 | 1.87 | 14.42 | 45.35 | 1.87 | 14.42 | 45.35 |
| 3 | 1.46 | 11.22 | 56.57 | 1.46 | 11.22 | 56.57 |
| 4 | 1.31 | 10.04 | 66.61 | 1.31 | 10.04 | 66.61 |
| 5 | 1.05 | 8.07 | 74.67 | 1.05 | 8.07 | 74.67 |
| 6 | 0.85 | 6.54 | 81.21 | 0.85 | 6.54 | 81.21 |
| 7 | 0.72 | 5.53 | 86.74 | 0.72 | 5.53 | 86.74 |
| 8 | 0.67 | 5.12 | 91.86 | | | |
| 9 | 0.45 | 3.42 | 95.29 | | | |
| 10 | 0.1 | 2.34 | 99.63 | | | |
| 11 | 0.03 | 0.25 | 100 | | | |

As can be seen in Table 1, the variance contribution rate of the first principal component can reach 30.93%, indicating that it contains most of the information of the selected factors. The variance contribution rate of other principal components is getting smaller, which means that the information of the selected factors is less and fewer. The first seven principal components are determined as the forecasting factors, the cumulative variance contribution rate of which can reach 86.74%. According to equation (1), these seven principal components are the linear combinations of eleven initial forecasting factors, including the Eastern Pacific Subtropical High Northern Boundary Position Index (in June last year), the Pacific Subtropical High Northern Boundary Position Index (in June last year), the Atlantic Meridional Mode SST Index (in July last year), etc. Then, the score coefficient matrix of principal components is shown in Table 2.

**Table 2.** Score coefficient matrix of principal components.

| Factor Type | Component | | | | | | |
|---|---|---|---|---|---|---|---|
| | 1 | 2 | 3 | 4 | 5 | 6 | 7 |
| Eastern Pacific Subtropical High Northern Boundary Position Index (Last June) | 0.569 | −0.048 | 0.046 | 0.019 | −0.101 | −0.054 | 0.189 |
| Pacific Subtropical High Northern Boundary Position Index (Last June) | 0.496 | 0.001 | −0.003 | 0.045 | 0.112 | −0.105 | −0.137 |
| Atlantic Meridional Mode Sea Surface Temperature Index (Last July) | −0.039 | −0.034 | 0.15 | −0.028 | 0.13 | 0.92 | 0.14 |
| Cold-Tongue EI Nino-Southern Oscillation Index (Last August) | 0.013 | 0.089 | 0.024 | −0.019 | −0.09 | 0.137 | 0.926 |
| Pacific Polar Vortex Intensity Index (Last September) | −0.267 | 0.551 | −0.004 | 0.276 | −0.158 | 0.27 | −0.299 |
| Northern Hemisphere Polar Vortex Central Longitude Index (Last September) | −0.015 | −0.083 | −0.031 | −0.13 | 0.93 | 0.135 | −0.086 |
| Eurasian Meridional Circulation Index (Last September) | 0.059 | 0.129 | 0.431 | 0.298 | 0.195 | −0.203 | 0.006 |
| Atlantic-European Polar Vortex Intensity Index (Last October) | 0.159 | 0.686 | −0.002 | −0.26 | 0.035 | −0.226 | 0.314 |
| NINO C Sea Surface Temperature Anomaly Index (Last September) | 0.007 | −0.046 | 0.823 | −0.12 | −0.058 | 0.186 | 0.031 |
| Pacific Subtropical High Northern Boundary Position Index (Last August) | 0.034 | −0.061 | −0.088 | 0.848 | −0.133 | 0.005 | −0.044 |
| Rainfall (Last year) | 0.06 | 0.13 | 0.301 | −0.103 | −0.468 | 0.635 | 0.121 |

### 4.2. Four Annual Runoff Forecasting Models

Seven forecasting factors determined by the PCA method are taken as input conditions of the MLR model, BPNN model, ENN model, and PSO-SVR model, respectively. In this paper, the trial and error method is adopted to compare the prediction results of different forecasting models and determine the optimal parameters used in the models. Taking the ENN model as an example, different combinations of the node number in the input layer and hidden layer are proposed to determine the optimal combination. As a result, the optimal node number of the input layer is seven, while the optimal that of the hidden layer is eight. Thus, the combination of node numbers in the ENN model is (seven, eight, one). Then, major structures of four forecasting models are listed in Table 3, such as the regression equation of the MLR model, as well as other parameter settings of the BPNN model (e.g., node number, maximum training time, learning rate), the ENN model (e.g., node number, maximum training time, learning rate), and the PSO-SVR model (e.g., population size, maximum iteration time, $C$, $\varepsilon$, $\sigma$), respectively.

**Table 3.** Major structures of four forecasting models.

| Model | Model Structure | | |
|---|---|---|---|
| MLR | | Regression Equation $y = 0.9402x + 236.98$ | |
| BPNN | Node Number (7, 8, 1) | Maximum Training Time 5000 | Learning Rate 0.75 |
| ENN | Node Number (7, 8, 1) | Maximum Training Time 3000 | Learning Rate 0.95 |
| PSO-SVR | Population Size 200 | Maximum Iteration Time 2000 | $(C, \varepsilon, \sigma)$ (5.6, 0.0003, 3.4) |

In order to compare the performances of the proposed MLR model, BPNN model, ENN model, and PSO-SVR model of the Dongbei Hydrological Station in the Ganjiang River Basin, the annual runoff time series predicted by four forecasting models are shown in Figure 6, and the comparison results of three evaluation indexes are illustrated in Table 4.

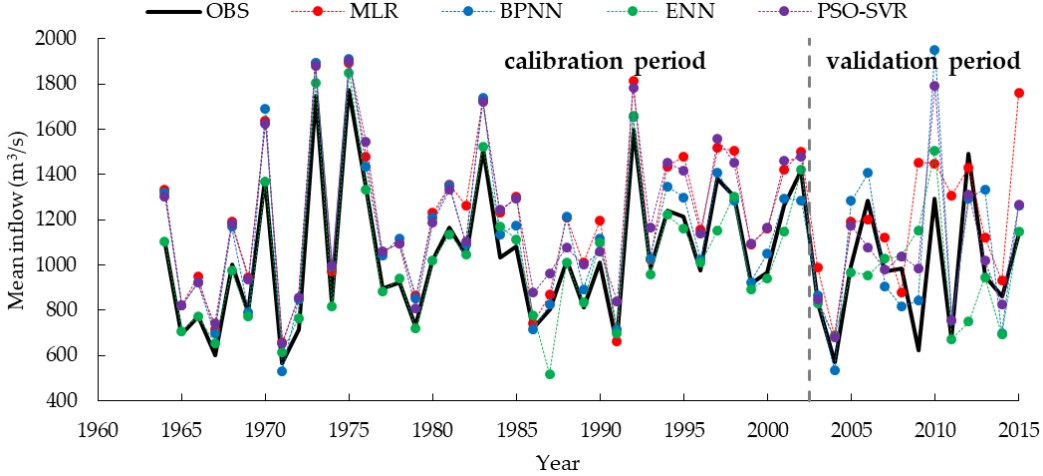

**Figure 6.** Annual runoff time series predicted by four forecasting models (note: OBS represents the observed value. MLR, BPNN, ENN, and PSO-SVR represent the predicted value of the MLR model, the BPNN model, the ENN model and the PSO-SVR model, respectively).

**Table 4.** Evaluation indexes of four runoff forecasting models (note: MARE, RMSE and QR represent mean absolute relative error, root mean squared error, and qualified rate, respectively).

| Model | Calibration Period | | | | Validation Period | | | |
|---|---|---|---|---|---|---|---|---|
| | MARE | RMSE | QR | NS | MARE | RMSE | QR | NS |
| MLR | 15.72% | 170.96 | 82.05% | 0.682 | 32.08% | 357.17 | 69.23% | −0.832 |
| BPNN | 10.55% | 127.93 | 94.87% | 0.822 | 19.60% | 256.07 | 69.23% | 0.058 |
| ENN | 4.63% | 73.64 | 97.44% | 0.941 | 17.81% | 282.32 | 76.92% | −0.144 |
| PSO-SVR | 15.99% | 165.03 | 82.05% | 0.704 | 15.73% | 202.50 | 84.62% | 0.411 |

As shown in the figure above, it is evident that the MLR model has the worst prediction performance whether runoff time series is in the calibration period or in the validation period, except that MARE of the PSO-SVR model is larger than that of the MLR model in the calibration period. By contrast, the ENN model has the best prediction performance in the calibration period, because this model has the smallest MARE, the smallest RMSE, the largest QR, and the largest NS, in terms of evaluation indexes. Likewise, the PSO-SVR model with the best prediction performance is glaringly obvious in the validation period, MARE, RMSE, QR, and NS of which are 15.73%, 202.50, 84.62%, and 0.411, respectively. In addition to these, it is obvious that the prediction performance of the BPNN model is superior to that of the MLR model, expect that QR values of both models are 69.23%.

### 4.3. Coupling Annual Runoff Forecasting Model

Aimed at proposing a coupling forecasting model by considering three forecasting models (i.e., BPNN model, ENN model, PSO-SVR model) that have different prediction performances, the historical annual runoff time series are divided into three hydrological years (i.e., wet year, normal year, dry year), and three coupling multi-model parameter equations based on the least-square method are introduced to train annual runoff time series in the calibration period from the aspect of hydrological years. Then, these coupling multi-model parameter equations are adopted to verify the annual runoff time series in the validation period. Coupling multi-model parameter equations are shown in Table 5.

**Table 5.** Coupling multi-model parameter equations corresponding to different hydrological years (note: $y$ means the annual runoff predicted by the coupling model. $x_1, x_2, x_3$ mean the annual runoff predicted by the BPNN model, the ENN model, and the PSO-SVR model, respectively).

| Hydrological Year | Coupling Multi-Model Parameter Equation |
|---|---|
| Wet Year | $y = 0.0253x_1 + 0.7938x_2 − 0.3046x_3 + 765.3838$ |
| Normal Year | $y = 0.0382x_1 + 0.4586x_2 + 0.303x_3 + 155.1607$ |
| Dry Year | $y = 0.1577x_1 − 0.0369x_2 + 0.6629x_3 + 49.6414$ |

Based on these coupling equations, the annual runoff time series predicted by the coupling model is shown in Figure 7, and the comparison results of three evaluation indexes are illustrated in Table 6. It is demonstrated that the prediction performance of annual runoff time series in the calibration period is better than that in the validation period, and the former predicted value is closer to the observed value than the latter one. As far as the calibration period is concerned, MARE, RMSE QR, and NS of the coupling model are 2.92%, 42.13, 100%, and 0.980, respectively, while that of the ENN model are 4.63%, 73.64, 97.44%, and 0.941, respectively. When it comes to the validation period, MARE, RMSE, QR, and NS of the coupling model are 8.06%, 157.90, 84.62%, and 0.642, respectively, while that of the PSO-SVR model are 15.73%, 202.50, 84.62%, and 0.411, respectively. Thus, it is concluded that this coupling model has better model performances than any single forecasting model (i.e., BPNN model, ENN model, PSO-SVR model) mentioned above, because this coupling model has the smaller MARE, the smaller RMSE, the larger QR, and the larger NS, in terms of evaluation indexes. However, it can be clearly seen that the least-square method is greatly affected by the disturbance of the outlier (the highest

blue peak in the validation period in Figure 7) when modifying the predicted values, which is closely related to its correction strategy. The least-square method takes the distance as the measure, and the parameters of the fitting function are obtained by the least-square sum of errors, which will enlarge the influence of the large error to this method.

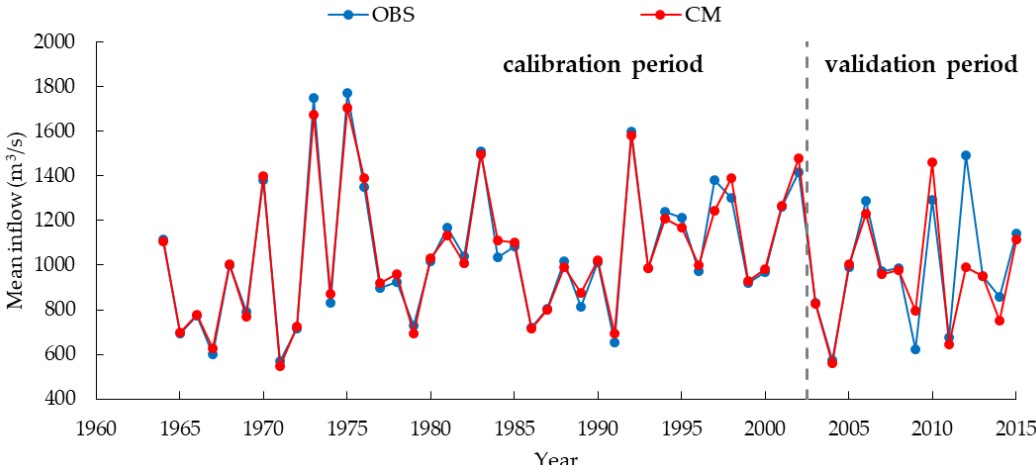

**Figure 7.** Prediction results of the coupling model (note: OBS represents the observed value, CM represents the predicted value of the coupling model).

**Table 6.** Evaluation indexes of the coupling model.

| Model | Calibration Period | | | | Validation Period | | | |
|---|---|---|---|---|---|---|---|---|
| | MARE | RMSE | QR | NS | MARE | RMSE | QR | NS |
| Coupling model | 2.92% | 42.13 | 100% | 0.980 | 8.06% | 157.90 | 84.62% | 0.642 |

### 4.4. Modified Coupling Annual Runoff Forecasting Model

Based on the observed values and the predicted values of the coupling model from 1964 to 2002, a modified function is proposed by means of residual error correction, and the residual correction function equation can be expressed as follows:

$$x_i = -0.0014y_i - 5.6197. \tag{20}$$

Then, the prediction sequences of the coupling model $y_i$ are taken into the model correction function, and the residual error sequences $x_i$ can be obtained. Subsequently, modified annual runoff sequences from 2003 to 2015 can be calculated, and comparisons of annual runoff sequences predicted by the coupling forecasting model and the modified coupling forecasting model are illustrated in Table 7.

As shown in the table above, MARE of the coupling model is 8.06%, while that of the modified coupling model is 7.99%. RMSE of the coupling model is 157.90, while that of the modified coupling model is 156.69. Moreover, QR of the coupling model is 0.642, while that of the modified coupling model is 0.647. Therefore, the modified coupling model can further improve the prediction performance of annual runoff time series in the validation period, with the help of the residual error correction technique.

**Table 7.** Comparisons of annual runoff sequences predicted by the coupling model and the modified coupling model (note: units of the observed value and the predicted value are m$^3$/s).

| Year | Observed Value | Coupling Model | | | Modified Coupling Model | | |
|---|---|---|---|---|---|---|---|
| | | Predicted Value | ARE | Qualified | Predicted Value | ARE | Qualified |
| 2003 | 831.37 | 825.93 | 0.65% | Yes | 832.71 | 0.16% | Yes |
| 2004 | 571.85 | 559.44 | 2.17% | Yes | 565.84 | 1.05% | Yes |
| 2005 | 992.82 | 1003.93 | 1.12% | Yes | 1010.95 | 1.83% | Yes |
| 2006 | 1287.04 | 1229.97 | 4.43% | Yes | 1237.31 | 3.86% | Yes |
| 2007 | 972.21 | 958.45 | 1.42% | Yes | 965.41 | 0.70% | Yes |
| 2008 | 986.38 | 977.66 | 0.88% | Yes | 984.64 | 0.18% | Yes |
| 2009 | 621.56 | 793.89 | 27.73% | No | 800.62 | 28.81% | No |
| 2010 | 1292.61 | 1462.52 | 13.14% | Yes | 1470.18 | 13.74% | Yes |
| 2011 | 674.28 | 644.03 | 4.49% | Yes | 650.55 | 3.52% | Yes |
| 2012 | 1491.01 | 993.14 | 33.39% | No | 1000.15 | 32.92% | No |
| 2013 | 949.76 | 948.97 | 0.08% | Yes | 955.91 | 0.65% | Yes |
| 2014 | 859.82 | 749.95 | 12.78% | Yes | 756.62 | 12.00% | Yes |
| 2015 | 1142.31 | 1113.25 | 2.54% | Yes | 1120.43 | 1.92% | Yes |

## 5. Conclusions and Discussions

For the sake of improving the forecasting accuracy of annual runoff time series, the principal component analysis (PCA) method is adopted to screen forecasting factors from rainfall, runoff, and 130 monitoring indexes. A modified coupling runoff forecasting model is proposed based on multiple linear regression (MLR), back propagation neural network (BPNN), Elman neural network (ENN), and particle swarm optimization-regression support vector machine (PSO-SVR), by means of multi-model information fusion and residual error correction in this paper. The main conclusions of this study are as follows:

Firstly, seven principal components are screened as key forecasting factors from the Eastern Pacific Subtropical High Northern Boundary Position Index (in June last year), the Pacific Subtropical High Northern Boundary Position Index (in June last year), the Atlantic Meridional Mode SST Index (in July last year), and other monitoring indexes, as well as annual rainfall.

Then, compared to the MLR model, the BPNN model, the ENN model, and the PSO-SVR model provide better prediction performances involving predicting annual runoff. In terms of a single forecasting model, the PSO-SVR model has the best prediction performances in the validation period, while the ENN model has the best prediction performances in the calibration period.

Subsequently, from the point of hydrological years (i.e., wet year, normal year, dry year), a coupling model is proposed by means of multi-model parameter equations by taking the advantages of three forecasting models (i.e., BPNN, ENN, PSO-SVR) into account. MARE, RMSE, QR, and NS of the coupling model are 2.92%, 42.13, 100%, and 0.980, while that of the ENN model are 4.63%, 73.64, 97.44%, and 0.941 in the calibration period. MARE, RMSE, QR, and NS of the coupling model are 8.06%, 157.90, 84.62%, and 0.642, while that of the PSO-SVR model are 15.73%, 202.50, 84.62%, and 0.411 in the validation period.

Finally, the residual error correction technique is referenced to modify runoff sequences predicted by the coupling model in the validation period. Compared to the coupling model, the modified coupling model has the smaller MARE, the smaller RMSE, and the larger NS.

In conclusion, the modified coupling method proposed in this paper can be applied to the Ganjiang River Basin to improve prediction performances of annual runoff sequences. It would also play an important role in providing a significant improvement of runoff forecasting in other similar river basins, especially by means of multi-model information fusion techniques to combine the advantages of different forecasting models, which is an important innovation in this paper. Besides, the residual error correction method in the modified coupling model is another feature in this study that would further improve the performance of the predicted annual runoff in the validation period, based on the least-square sum of errors. Nevertheless, there is still some room for perfection in this study,

for example the number of nodes in the input layer and the hidden layer is determined by the trial and error method, rather than the optimization method, which may reduce the calculation efficiency. Using optimization algorithms to determine the number of nodes and weights of neural networks may be the trend of future development. The least-square method is not always a good correction strategy when there are outliers in the predicted results, that is, the error between the predicted value and the observed value is large in the coupled model. Mean absolute error method and smoothed mean absolute error method might be alternative options for data fitting. Therefore, making full use of the advantages of different models to achieve the optimality of the coupled model remains a challenge in the forecasting runoff model, which will also be the focus of future research.

**Author Contributions:** P.S., C.W., and W.L. conducted the research and designed and wrote the paper; J.S., L.K., and Z.N. helped to revise the paper; X.L. and H.W. gave the comments. All authors have read and agreed to the published version of the manuscript.

**Funding:** This research was funded by the National Natural Science Foundation of China (51709275), the Young Elite Scientists Sponsorship Program by CAST (2019QNRC001), the Fundamental Research Funds of IWHR (WR0145B012020), and the National Key Research and Development Plan of China (2018YFC0407405).

**Acknowledgments:** The anonymous reviewers and the editor are thanked for providing insightful and detailed reviews that greatly improved the manuscript.

**Conflicts of Interest:** The authors declare no conflict of interest.

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
