# Peer review of "Annual Runoff Forecasting Based on Multi-Model Information Fusion and Residual Error Correction in the Ganjiang River Basin"

_water, doi:10.3390/w12082086_

Round 1

Reviewer 1 Report

In the manuscript entitled Annual Runoff Forecasting Based on Multi-Model Information Fusion and Residual Error Correction in the Ganjiang River Basin, the authors was presented the results obtained from four prognostic models: model based on multiple linear regression (MLR), back propagation neural network (BPNN), Elman neural network (ENN) and particle swarm optimization-support vector machine for regression (PSO-SVR). These models were used to forecast annual runoff time series. Aimed at improving the prediction accuracy of annual runoff time series, principal components analysis method is adopted to reduce the complexity of forecasting factors. In addition, information for wet year, normal year, dry year has been introduced in BPNN, ENN and PSO-SVR models. In order to further improve the accuracy of the predicted annual runoff time series in the  verification period modified coupling forecasting model, residual error correction was improved.

The manuscript is focused on the presentation of results mainly, with a very general description of the models used. Model quality was assessed using: mean absolute percentage error (MAPE), root mean squared error (RMSE) and qualified rate (QR). It was found that the PSO-SVR model has the best prediction performances in the validation period, while the ENN model has the best prediction performances in the calibration period. Modified coupling method proposed in this manuscript, improves prediction performances of annual runoff sequences.

Notes - in my opinion the important part of the manuscript "discussion" is missing, which makes it difficult to assess the significance of the results obtained, especially in relation to the parameters used in the models. The "Conclusions" part should be replaced with "Summary". The issues raised in item 5 are replicates of the results evaluation in items 4 "Results" so in this part of the manuscript there are no right conclusions.

The manuscript is acceptable and could be published in the journal Water. Comments on the lack of a "discussion" point and conclusions, would improve the readability of the manuscript, but their disregard by the authors does not negatively affect on the quality of the manuscript.

Author Response

Dear Reviewer:

Thank you for your anonymous comments concerning our manuscript entitled “Annual Runoff Forecasting Based on Multi-Model Information Fusion and Residual Error Correction in the Ganjiang River Basin” (No. water-865241). Those comments are all valuable and very helpful for revising and improving our paper, as well as the important guiding significance to our researches. We have studied the comments carefully and made revisions accordingly, which we hope meet with your approval. Revised portion are marked in red in the paper and the detailed corrections are listed below point by point.

Reviewer 2 Report

A well prepared paper on the forecasting of annual runoff through the use and combination of different methods and techniques. Below are some concerns that need to be addressed before the manuscript can be accepted.

The abstract needs to be completed with some insights about the results achieved and main conclusions drawn.

Overall, the introduction is well approached. It contains a short initial paragraph that is sufficient to introduce the pproblem. Then, there is a detailed review overviewing previous approach for hydrological forecasting.

However, the last paragraph of the section needs to be strenghtened. What are the contributions of the manuscript and how this research improves previous approaches? This is unclear. I don't think that fusing a series of methods is something new in the field, so what is really original and innovative here?

Please, include the year of publication when making explicit citations.

Please, avoid informal expressions such as "relatively speaking".

Subsection 3.5. What about the Nash-Sutcliffe coefficient? It is a widely used goodness-of-fit measure in hydrological studies. 

Section 4 needs to be re-approached to move from just "results" to "results and discussion". The authors need to reason and justify the results achieved from a physical point of view. For instance, what about the outlier (the highest blue peak in Figure 7 validation period) that is preventing the model from having higher accuracy?

The conclusions need to be rewritten. In its current condition, this section is a mere summary of the manuscript. Only lines 481-484 are relevant for a conclusions section. The authors have to focus on the main findinds of the investigation and their implications for the fields of hydrology and water policy and decision-making. The limitations of the research should be pointed out as well. Finally, some future lines of action to overcome such limitations must be included as further research to be developed.

Author Response

(The authors gave the same response as above.)
